# GENERALIZABLE DEEP RL-BASED TSP SOLVER VIA APPROXIMATE INVARIANCE

## ABSTRACT

Recently, deep reinforcement learning (DRL) has shown promising results for learning fast heuristics to solve traveling salesman problems (TSP). Meanwhile, most existing state-of-the-art (SOTA) DRL methods yield solvers that do not generalize well on TSP instances larger than those seen during training. However, such generalization ability is crucial in practice since training on large instances is impractical. To tackle this issue, we propose a novel DRL method, called $TS^3$, which is designed to enforce a variety of (possibly approximate) invariances to promote the generalizability of the learned solver. More specifically, $TS^3$ applies a modified policy gradient algorithm enhanced with data augmentation to train a Transformer-based model to select the next city to visit among the k-nearest neighbors of the last visited city by integrating a local view and global view of a TSP instance. To further validate the capability of $TS^3$, we also propose its combination with Monte-Carlo Tree Search. Abundant experiments on random TSP and TSPLIB instances demonstrate that our propositions achieve a dominant performance when generalizing to large-sized TSPs.

## 1 INTRODUCTION

Among all combinatorial optimization problems, Traveling Salesman Problem (TSP) is arguably one of the most popular thanks notably to the simplicity of its formulation and its wide application range, such as logistics (Madani et al., 2020), electronic design automation (Alkaya & Duman, 2013), or bioinformatics (Matai et al., 2010). In this problem, given a graph, the goal is to find a shortest tour that visits all the nodes exactly once while returning to a starting node. Due to its NP-hard nature, exact algorithms are impracticable to solve large-sized instances, which motivates the active development of approximate heuristic methods. Although state-of-the-art (SOTA) heuristic methods, such as LKH3 (Helsgaun, 2009; 2017) have been designed to provide high-quality solution for large TSP instances faster than exact methods, they are still too computationally costly.

To obtain faster heuristics, researchers have started to actively explore the exploitation of deep learning, and especially deep reinforcement learning (DRL), to design TSP solvers, e.g., Attention Model (Kool et al., 2019), or PointerFormer (Jin et al., 2023), which are generally constructive (i.e., they generate the solution by iteratively select the next node to visit from the last visited one). Though this approach shows promising results, the proposed models do not generally reveal good generalization ability (Joshi et al., 2020). Indeed, most work can only achieve good performance on TSP instances whose size is close to the training instance sizes. Thus, models trained on small-sized instances are incapable of generating a satisfying solution on large-sized instances, which could only be tackled by a model trained on large-sized instances. However, such training would cost a large amount of time and computational resource, making it impractical.

Our work aims at better understanding how approximate invariance can promote cross-size generalization (omit cross-size in the rest of this article) in a DRL-based TSP solver. Figure 1 (Left) shows the distribution of the rank of the next node among the nearest neighbors of a current node in an "optimal" tour for different TSP instances. This figure suggests that an optimal tour can generally still be obtained by only focusing on the k-nearest neighbors (k-NNs) of the last visited node. Based on this observation, we propose (1) to directly restrict the action space of a DRL agent to the k-NNs of the last visited node and (2) to provide two views to this agent as its state: a local view focused on the k-NNs and a global view including all the unvisited nodes. The first idea simplifies

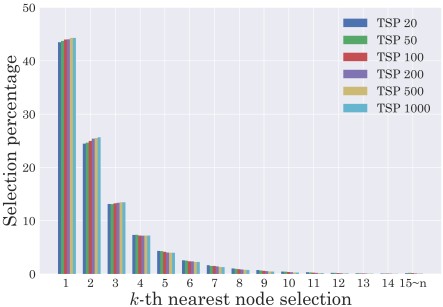 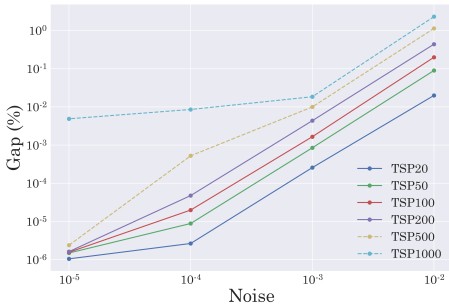

Figure 1: Empirical regularity observed on random TSP instances (results averaged over 1000 instances per TSP size). (Left) Distribution of the rank of the next node to visit from a node in a solution tour among the nearest neighbors of the latter node. (Right) Percentage change (gap) of the quality of solutions of instances perturbed by random noise. Note that the solutions here are produced by LKH3, which can output exact optimal solutions for small-sized instances and near-optimal solutions for large-sized instances. The dashlines in the right figure denotes the variation might be caused by the sub-optimality of LKH3, rather than the random noise.

the decision-making problem by directly choosing among the most probable nodes, while the second idea allows the local and global views to be processed separately, which enables more efficient invariant preprocessing (e.g., scaling) of the k-NNs. Interestingly, the first idea can be understood as approximate invariance since focusing on the k-NNs amounts to expressing the independence with respect to the nodes that are farther away. Furthermore, Figure 1 (Right) shows how much the quality of an "optimal" solution changes (i.e., gap) when an instance is perturbed by random noise (i.e., all node positions are changed by small random noises). Note that while the solutions obtained by LKH3 are optimal with high probability, they are generally suboptimal for TSP500 and TSP1000, which indicates that the corresponding curves in dashed lines are less reliable. This figure shows that small perturbations introduce small gaps, which can be regarded as exploiting approximate invariance. This observation motivates us to apply various invariant transformations (e.g., rotation or reflection) with random perturbation to a TSP instance to generate many instances sharing similar optimal solutions.

To make these ideas operational for exploiting (approximate) invariance, we design a novel Transformer-based model for solving TSP and propose a modification of the REINFORCE algorithm to train it. In addition, we also combine our trained model with Monte-Carlo Tree Search (MCTS). Our contributions are summarized below.

- We propose a novel Transformer-based architecture combining local and global information enforcing approximate invariance in the policy. The local view includes the $k$-NN graph centered at the last visited node. The global view includes all nodes dependent on the optimal solution.

- We propose a novel training method exploiting approximate invariance by involving exact-invariant operations (e.g. rotation) and approximate-invariant operation (e.g. noisy perturbation).

- In addition, to demonstrate the quality of $\texttt{TS}^3$, we formulate a generic approach $\texttt{TS}^4$ to derive a heatmap from a constructive method to be used in MCTS.

- We perform comprehensive experiments to evaluate the generalizability of our method, compared with many other available methods. We also conduct an ablation study and sensitivity experiments to validate the positive effects of our design decisions.

## 2 RELATED WORK

In this section, we discuss related work and emphasize how it differs from our proposition. Here we only include those work having strong relationship with us, and more additional work over TSP can be found in Appendix B.

**Transformer Structure.** The Transformer (Vaswani et al., 2017) model has inspired multiple architectures proposed for solving TSP. Notably, Kool et al. (2019) design an attention model (AM) using attention layers, while Bresson & Laurent (2021) show that the original Transformer model works well on small-sized TSP. Jin et al. (2023) create a multi-pointer network, called Pointerformer, to aggregate information from all nodes. In contrast, our proposition, which exploits approximate invariance by notably focusing on nearest neighbors, uses attention layers to process and use local and global views when selecting the next node to visit.

**Local and Global Information.** Jiang et al. (2023) proposes a MVGCL to leverage local information using kNN on the whole graph for learning representative features by contrastive learning. This kind of $k$-NN usage is applied on the complete graph, which is different from the $k$-NN during tour construction. Gao et al. (2023) explores a local view based on a $k$-NN during construction and combines the outputs of a local policy and a global policy. Our design uses only one local policy, which enforces a $k$-NN approximate invariance over the policy, and the aggregation of the local and global views is performed in the embedding space.

**Data Augmentation and Invariance.** Kwon et al. (2020) achieve with POMO a great performance gain by applying data augmentation to handle multiple trajectories at both the training phase and the test phase. Ouyang et al. (2021) demonstrate with eMAGIC that exploiting invariance (via data augmentation and preprocessing) can help generalization. Kim et al. (2022) further develops a formal algorithm to learn invariant policy for combinatorial optimizations by rotation augmentation. In addition to exact invariance, our method also exploits approximate invariance both when applying data augmentation and in the architecture design to further enhance generalizability.

**Monte-Carlo Tree Search.** Fu et al. (2021) train a specific model to predict probability heatmaps of TSP instances, and then utilize MCTS to optimize the solutions. Their MCTS samples a k-opt local move according to a probability heatmap over all edges, which is updated by the performance of sampled local moves. Our proposition $\texttt{TS}^4$ follows the same MCTS scheme, but we construct the initial solutions and the probability heatmaps from our model.

## 3 BACKGROUND

We first specify some mathematical notations used in the rest of this article. Basic knowledge of TSP and DRL is then recalled, and the problem is finally formalized.

**Notations.** For any positive integer $n \in \mathbb{N}$, $[n]$ denotes the set $\{1, 2, \cdots, n\}$. Set $\mathbb{S}_n$ represents the set of all permutations of $[n]$, where a permutation is denoted $\boldsymbol{\sigma} = (\sigma_1, \cdots, \sigma_n)$. By extension, for any finite set $\mathcal{X} = \{x_1, x_2, \cdots, x_n\}$, $\boldsymbol{\sigma}(\mathcal{X}) = (x_{\sigma_1}, \cdots, x_{\sigma_n})$ denotes a permutation of the elements of $\mathcal{X}$. For any finite set $\mathcal{X}$, $\Delta(\mathcal{X})$ denotes the set of probability distributions over $\mathcal{X}$.

### 3.1 EUCLIDEAN TSP

A Euclidean TSP instance can be described as a set $\mathcal{C} = \{c_1, c_2, \ldots, c_n\}$ of cities, where each city $c_i$ has coordinates $(x_i, y_i) \in [0, 1]^2$. This set induces a graph $\mathcal{G} = (\mathcal{C}, \mathcal{E})$, where $\mathcal{E} = \{e_{i,j} \mid i, j \in [n]\}$ denotes the set of edges, and each edge $e_{i,j} = \{c_i, c_j\}$ has a cost defined as the Euclidean distance between cities $c_i$ and $c_j$: $D(e_{i,j}) = \sqrt{(x_i - x_j)^2 + (y_i - y_j)^2}$. In a TSP instance, the salesman has to visit all the cities exactly once and return to its initial city, while minimizing the travelled distance. Formally, such a sequence of city visits is called a *tour*, which can be encoded as a permutation $\boldsymbol{\sigma}$ of the city indices. The length of a tour can be calculated as:

$$L_{\mathcal{C}}(\boldsymbol{\sigma}) = D(c_{\sigma_n}, c_{\sigma_1}) + \sum_{i=1}^{n-1} D(c_{\sigma_i}, c_{\sigma_{i+1}}). \tag{1}$$

Therefore, solving a TSP instance $\mathcal{C}$ amounts to finding an optimal tour $\boldsymbol{\sigma}^*$ with minimal length.

### 3.2 MDP AND RL

Reinforcement learning is based on the Markov Decision Process (MDP) model, which can be described by a tuple $\mathcal{M} = (\mathcal{S}, \mathcal{A}, P, r, \gamma, d_0)$, where $\mathcal{S}$ and $\mathcal{A}$ represent a state space and an action

space respectively, $P : \mathcal{S} \times \mathcal{A} \to \Delta(\mathcal{S})$ is a transition function, $r : \mathcal{S} \times \mathcal{A} \mapsto \mathbb{R}$ is a reward function, $\gamma \in (0, 1]$ is a discount factor, and $d_0 \in \Delta(\mathcal{S})$ is an initial state distribution. A (stochastic) policy $\pi : \mathcal{S} \to \Delta(\mathcal{A})$ selects an action stochastically given a current state.

The objective of reinforcement learning is to find an optimal policy $\pi^*$ to maximize the total expected discounted rewards (assuming episodic problems with horizon $T$):

$$J(\pi) = \mathbb{E}_{s_0 \sim d_0, a_t \sim \pi(\cdot|s_t), s_{t+1} \sim P(s_t, a_t, \cdot)} [G] \quad \text{where} \quad G = \sum_{t=0}^{T-1} \gamma^t r(s_t, a_t), \quad (2)$$

where value $G$ is the so-called *episodic return*.

### 3.3 FORMALIZING TSP AS AN RL PROBLEM

Constructive methods solve a combinatorial optimization problem by generating a whole solution step by step. In Euclidean TSP, such methods start from an initial node, and repeatedly select the next node to visit, until all nodes have been visited. This iterative process can be formalized as an MDP. At time step $t$, a state $s_t = (\boldsymbol{\sigma}(\mathcal{V}_t), \mathcal{C})$ consists of a current partial tour $\boldsymbol{\sigma}(\mathcal{V}_t) = (c_{\sigma_1}, \cdots, c_{\sigma_t})$ (i.e., permutation of already-visited nodes in $\mathcal{V}_t \subset \mathcal{C}$) and a TSP instance $\mathcal{C}$. Thus, the state space is defined as $\mathcal{S} = \{(\boldsymbol{\sigma}(\mathcal{V}_t), \mathcal{C}) \mid t \in [n], \mathcal{V}_t \subseteq \mathcal{C}, |\mathcal{V}_t| = t, \boldsymbol{\sigma} \in \mathbb{S}_t\}$. In a state $s_t$, an action is any node $c_i$ in the set of unvisited nodes $\mathcal{U}_t = \mathcal{C} \setminus \mathcal{V}_t$. Thus, the action space is simply $\mathcal{A} = \mathcal{C}$, but in a state $s_t$, only actions in $\mathcal{U}_t$ are allowed. In this MDP, transitions are deterministic: selecting $c_i \in \mathcal{U}_t$ in state $s = (\boldsymbol{\sigma}(\mathcal{V}_t), \mathcal{C})$ leads to a new state where $c_i$ is removed from $\mathcal{U}_t$ and appended to current partial tour $\boldsymbol{\sigma}(\mathcal{V}_t)$. The reward here is simply the negative cost of the newly-added edge $e_{\sigma_t, i}$. Thus, with the discount factor set to $\gamma = 1$, episodic returns equal negative tour distances. By maximizing the expected returns, we learn a policy to minimize the tour length, ensuring the consistency between the objective of MDP and the objective of TSP.

As noticed by Kool et al. (2019), the optimal choice for the next city to visit is independent of the intermediate cities visited between the first visited node $c_{\sigma_1}$ and the last visited node $c_{\sigma_t}$, although they still use the previously-defined state space $\mathcal{S}$ as input since their encoder depends on the whole instance $\mathcal{C}$. Building on this idea, Ouyang et al. (2021) instead reduce the state space to $\mathcal{S} = \{(c_{\sigma_1}, c_{\sigma_t}, \mathcal{U}_t) \mid t \in [n], \mathcal{V}_t \subseteq \mathcal{C}, |\mathcal{V}_t| = t, \mathcal{U}_t = \mathcal{C} \setminus \mathcal{V}_t, \boldsymbol{\sigma} \in \mathbb{S}_t\}$, allowing the embeddings of unvisited nodes to be independent of visited ones. In this paper, we also use this reduced state space.

## 4 ARCHITECTURE

Our proposed deep RL method trains a differentiable architecture called $\text{TS}^2$ (**T**ransformer **S**tructured **T**SP **S**olver). It is composed of three components: *local encoder*, *global encoder*, and *decoder*. Figure 2 visualizes the model architecture, whose details are presented in the following paragraphs.

As explained in Section 1, an RL agent can be viewed as approximately invariant with respect to the $k$-NNs of the last visited node based on observations. Following this idea, we reduce the action set $\mathcal{U}_t$ to a set $\mathcal{U}_t^{\text{knn}}$ containing the $k$-NN of the last visited node. To focus on local information, the *local encoder* processes as inputs a partial state, which we call *local k-NNs*, defined as $(c_{\sigma_t}, \mathcal{U}_t^{\text{knn}})$. Since choosing the next node to visit only based on local information may be insufficient, we also introduce a global encoder. The *global encoder*, receiving the whole state $s = (c_{\sigma_1}, c_{\sigma_t}, \mathcal{U}_t)$ as inputs, is designed to compensate for this information loss. We denote nodes in the $k$-NNs as $\mathcal{U}_t^{\text{knn}} = \{c_{\varsigma_1}, \cdots, c_{\varsigma_k}\}$. The *decoder* takes as inputs the embeddings obtained from the two encoders and selects the next node to visit from $\mathcal{U}_t^{\text{knn}}$. Before explaining in details those components, we recall the definition of attention layer, which is the basic building blocks used in both encoders and decoder.

**Attention Layer.** Vaswani et al. (2017) propose a well-known encoder architecture built on attention layers. Each attention layer is equipped with residual mechanism and is composed of two key components: Multi-Head Attention (MHA) Layer and Feed Forward (FF) layer. Given the input embeddings $\boldsymbol{E}_{\text{enc}}^{(\ell)}$ of the $\ell$-th layer and embeddings $\boldsymbol{E_Q}, \boldsymbol{E_K}, \boldsymbol{E_V}$, the MHA layer and the FF layer

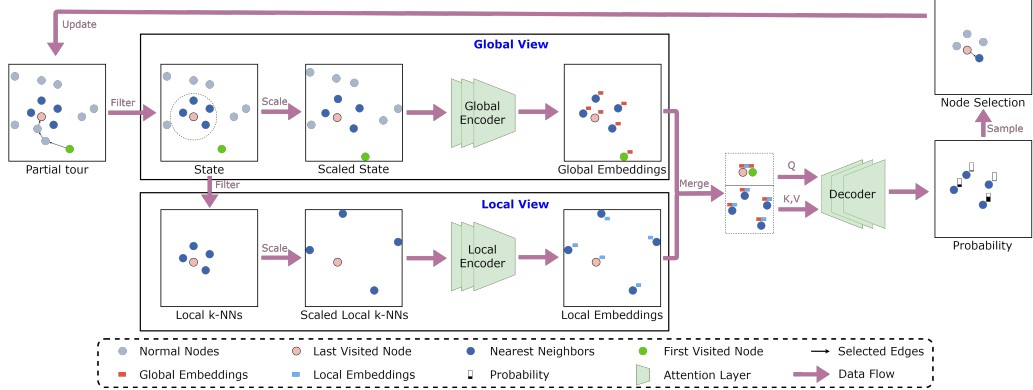

Figure 2: The overall architecture of $\text{TS}^2$. The input state is processed by the *local encoder* and the *global encoder*. The *local encoder*, aimed to provide representative local information, takes the scaled *local k-NNs* as inputs. The *global encoder*, aimed to compensate the information loss of the *local encoder*, take the scaled state as inputs. The *decoder* processes the merged information from the two encoders and outputs a final probability. The $\text{TS}^2$ samples a node to visit by the output probabilities and updates the partial tour in an auto-regressive manner, until a complete feasible tour is constructed. As for the first visited node, a learnable start placeholder is applied to find the suitable alternative. Best viewed in colors.

process the information as follows:

$$\boldsymbol{Q} = \texttt{Linear}(\boldsymbol{E_Q}), \quad \boldsymbol{K} = \texttt{Linear}(\boldsymbol{E_K}), \quad \boldsymbol{V} = \texttt{Linear}(\boldsymbol{E_V}) \tag{3}$$

$$\boldsymbol{E}_{\text{att}}^{(\ell+1)} = \texttt{BN}\left(\boldsymbol{E}_{\text{enc}}^{(\ell)} + \texttt{Softmax}\left(\frac{\boldsymbol{QK}^\top}{\sqrt{d_{\boldsymbol{K}}}}\right)\boldsymbol{V}\right), \tag{4}$$

$$\boldsymbol{E}_{\text{enc}}^{(\ell+1)} = \texttt{BN}\left(\boldsymbol{E}_{\text{att}}^{(\ell+1)} + \texttt{FF}\left(\boldsymbol{E}_{\text{att}}^{(\ell+1)}\right)\right), \tag{5}$$

where $\boldsymbol{Q}, \boldsymbol{K}, \boldsymbol{V}$, called queries, keys and values, are calculated by a linear layer from $\boldsymbol{E_Q}, \boldsymbol{E_K}, \boldsymbol{E_V}$ respectively. Scaling factor $d_{\boldsymbol{K}}$ is the feature dimension of the keys to normalize the inputs $\boldsymbol{QK}^\top$ of the softmax function. The outputs of the attention layer can be viewed as information aggregation for $\boldsymbol{Q}$ from $\boldsymbol{K}, \boldsymbol{V}$. The aggregated information for $\boldsymbol{Q}$ is directly added by the inputs $\boldsymbol{E}_{\text{enc}}^{(\ell)}$ following the residual mechanism. To make the two terms consistent, it is common to set $\boldsymbol{E_Q} = \boldsymbol{E}_{\text{enc}}^{(\ell)}$ when applying the attention layer.

**Local Encoder.** Guided by approximate invariance, the *local encoder* processes as inputs the *local k-NNs* $(c_{\sigma_t}, \mathcal{U}_t^{\text{knn}})$. This enables the *local encoder* to process a small fixed-sized group of nodes extracted from the original state. However, the density of nodes increases as the size of TSP increases. Higher density results in shorter distances between nodes, which makes it hard to generalize from small-sized TSP instances to large-sized TSP instances. To maintain the approximate invariance with respect to local information among different TSP sizes, $\text{TS}^2$ scales the coordinates of the nodes in the *local k-NNs* to a unit square $[0, 1]^2$. The scaled coordinates of these nodes are then processed as inputs by a linear embedding layer followed by multiple attention layers shown as follows:

$$\boldsymbol{E}^{(1)} = \texttt{Linear}\left(\texttt{Scale}(c_{\sigma_t}, \mathcal{U}_t^{\text{knn}})\right), \tag{6}$$

$$\boldsymbol{E}^{(\ell+1)} = \texttt{Attention}(\boldsymbol{E}^{(\ell)}, \boldsymbol{E_Q} = \boldsymbol{E_K} = \boldsymbol{E_V} = \boldsymbol{E}^{(\ell)}). \tag{7}$$

We call the outputs of the *local encoder* the *local embeddings* $\boldsymbol{E}^{\text{loc}} = (\boldsymbol{E}_{\sigma_t}^{\text{loc}}, \boldsymbol{E}_{\varsigma_1}^{\text{loc}}, \cdots, \boldsymbol{E}_{\varsigma_k}^{\text{loc}})$ (assuming $\mathcal{U}_t^{\text{knn}} = \{c_{\varsigma_1}, \cdots, c_{\varsigma_k}\}$), where $\boldsymbol{E}_\sigma$ is abused to represent the embeddings corresponding to node $c_\sigma$.

**Global Encoder.** The *global encoder* takes the whole state $s = (c_{\sigma_1}, c_{\sigma_t}, \mathcal{U}_t)$ as inputs. Similar to the *local encoder*, the *global encoder* also scales the coordinates and processes them with a linear embedding layer followed by multiple attention layers. Since the final action space is reduced to

$\mathcal{U}_t^{\text{knn}}$, the *global encoder* only provides the embeddings for $c_{\sigma_1}$, $c_{\sigma_t}$ and nodes in $\mathcal{U}_t^{\text{knn}}$ to improve computation efficiency. To achieve this, the queries are reduced to the embeddings corresponding to these nodes, which leads to the absence of the embeddings corresponding to $\mathcal{U}_t \setminus \mathcal{U}_t^{\text{knn}}$ in the layer outputs. However, if we process the embeddings same as in the *local encoder* where $\boldsymbol{E_K} = \boldsymbol{E_V} = \boldsymbol{E}^{(\ell)}$, the *global encoder* no longer aggregates the information from embeddings corresponding to $\mathcal{U}_t \setminus \mathcal{U}_t^{\text{knn}}$, because they are not provided in the layer outputs $\boldsymbol{E}^{(\ell)}$. To address this problem, we reuse the embeddings from the linear embedding layer to be keys and values for the attention layers. Formally, the whole state $s = (c_{\sigma_1}, c_{\sigma_t}, \mathcal{U}_t)$ is processed by the *global encoder* as follows:

$$\boldsymbol{E}^{(1)} = \texttt{Linear}\left(\texttt{Scale}(c_{\sigma_1}, c_{\sigma_t}, \mathcal{U}_t)\right), \tag{8}$$

$$\boldsymbol{E}^{(\ell+1)} = \texttt{Attention}(\boldsymbol{E}^{(\ell)}, \boldsymbol{E_Q} = (\boldsymbol{E}^{(\ell)}_{\sigma_1}, \boldsymbol{E}^{(\ell)}_{\sigma_t}, \boldsymbol{E}^{(\ell)}_{\varsigma_1}, \cdots, \boldsymbol{E}^{(\ell)}_{\varsigma_k}), \boldsymbol{E_K} = \boldsymbol{E_V} = \boldsymbol{E}^{(1)}). \tag{9}$$

The *global encoder* provides the *global embeddings* $\boldsymbol{E}^{\text{glo}} = (\boldsymbol{E}^{\text{glo}}_{\sigma_1}, \boldsymbol{E}^{\text{glo}}_{\sigma_t}, \boldsymbol{E}^{\text{glo}}_{\varsigma_1}, \cdots, \boldsymbol{E}^{\text{glo}}_{\varsigma_k})$.

**Decoder.** The *decoder* merges the local embeddings and the global embeddings, and outputs the final probabilities for node selection by a linear embedding layer followed by multiple attention layers. The *decoder* concatenates the *local embeddings* and the *global embeddings* corresponding to each node. Specifically, the *global embeddings* of the first visited node are directly concatenated with the embeddings of the last visited node. Then taking the concatenated embeddings of the first/last visited nodes as queries and the remaining concatenated embeddings as keys and values, the *decoder* applies attention layers.

$$\boldsymbol{E}^{(1)} = \texttt{Linear}\left(\texttt{Concat}(\boldsymbol{E}^{\text{loc}}_{\sigma_t}, \boldsymbol{E}^{\text{glo}}_{\sigma_t}, \boldsymbol{E}^{\text{glo}}_{\sigma_1})\right), \tag{10}$$

$$\boldsymbol{E}_{\text{knn}} = \texttt{Linear}\left(\left\{\texttt{Concat}(\boldsymbol{E}^{\text{loc}}_{\varsigma_i}, \boldsymbol{E}^{\text{glo}}_{\varsigma_i})\right\}_{i=1}^k\right), \tag{11}$$

$$\boldsymbol{E}^{(\ell+1)} = \texttt{Attention}(\boldsymbol{E}^{(\ell)}, \boldsymbol{E_Q} = \boldsymbol{E}^{(\ell)}, \boldsymbol{E_K} = \boldsymbol{E_V} = \boldsymbol{E}_{\text{knn}}), \tag{12}$$

and the probabilities for next node selection are computed by a softmax over the output embeddings $\boldsymbol{E}^{\text{dec}} = (\boldsymbol{E}^{\text{dec}}_{\varsigma_1}, \cdots, \boldsymbol{E}^{\text{dec}}_{\varsigma_k})$.

## 5 ALGORITHM

We propose $\texttt{TS}^3$ (for $\texttt{TS}^2$ + **T**ransformed **S**amples) by developing a deep RL algorithm to train our $\texttt{TS}^2$ model. We also develop $\texttt{TS}^4$ (for $\texttt{TS}^3$ + **T**ree **S**earch) by integrating the MCTS algorithm with our proposed deep RL methods. Appendix A.2 provides the algorithms written in pseudo-codes, which will be presented in this section.

### 5.1 ALGORITHM FOR $\texttt{TS}^3$

As explained in Section 1, the RL agent should be (approximately) invariant with respect to some transformations (e.g., Euclidean symmetry or random noise). Therefore, we utilize data augmentation to learn approximate invariance. Our algorithm is based on the REINFORCE algorithm, which is adopted in many previous papers on TSP.

**REINFORCE.** Assuming a parametric policy space $\{\pi_\theta \mid \theta \in \Theta\}$, the REINFORCE (Williams, 1992) algorithm optimizes eq. (2) via gradient ascent using the following gradient (so called *stochastic policy gradient* (Sutton et al., 1999)):

$$\nabla_\theta J(\pi_\theta) = \mathbb{E}_\eta\left[(G - B)\nabla_\theta \log p_\theta(\eta)\right] \tag{13}$$

where $\eta$ is the episodic trajectory with respect to distributions defined by $\pi_\theta, P, d_0$; $p_\theta(\eta)$ is the joint probability of the $\eta$; and $B$, called *baseline*, is to reduce the variance of the policy gradient.

Similar to Kool et al. (2019), a baseline model with parameters $\theta^{\text{BL}}$, which shares the same architecture with the train model with parameters $\theta$, is used to calculate baseline $B$.

**Data Augmentation.** The augmentation function $f \in \mathcal{F}$ includes operations of rotation, reflection, scaling, and noisy perturbation, with detailed presentations available in Appendix C.1. Given an original instance, we sample an augmentation function from $\mathcal{F}$ randomly for every generation of

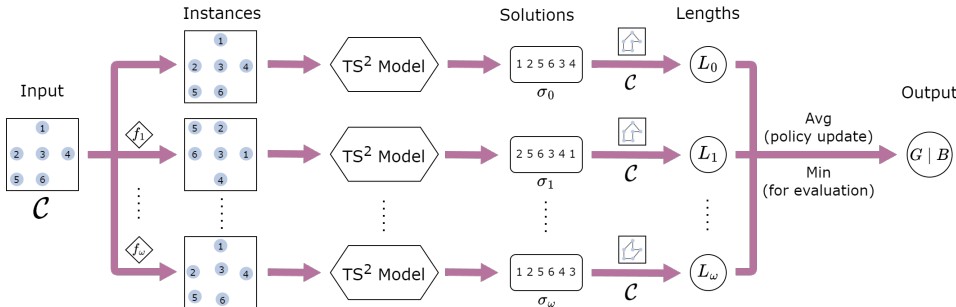

Figure 3: Data augmentation for $\text{TS}^3$. For a given TSP instance $\mathcal{C}$, more instances are generated by sampled augmentation functions. Solutions of all instances inferred by our $\text{TS}^2$ model are evaluated over $\mathcal{C}$. Average (resp. minimum) tour length represents the episodic return (resp. best result) of the input instance $\mathcal{C}$ for training (resp. evaluation). Notations on the figure are the same as in the text.

augmented instances. Augmented instances of number $\omega$ are generated for each original instance by $\omega$ sampled augmentation functions.

To generate the tours for TSP instances during training, the train model applies probabilistic sampling and the baseline model applies deterministic policy. In order to learn approximate invariance with respect to our defined data augmentation, tours for augmented instances are evaluated on there corresponding original instances. As discussed in Section 3.3, the episodic return and the baseline are derived by the negative tour lengths, with respect to tours generated by the train model and the baseline model respectively. Figure 3 visualizes the procedures described above. Formally, given a sampled TSP instance $\mathcal{C}$, our algorithm calculates:

$$G = L_{\mathcal{C}}\left(\boldsymbol{\sigma}(\pi_\theta, \mathcal{C})\right) + \omega \mathbb{E}_{f \in \mathcal{F}}\left[L_{\mathcal{C}}\left(\boldsymbol{\sigma}(\pi_\theta, f(\mathcal{C}))\right)\right], \tag{14}$$

$$B = L_{\mathcal{C}}\left(\boldsymbol{\sigma}(\mu_{\theta^{\text{BL}}}, \mathcal{C})\right) + \omega \mathbb{E}_{f \in \mathcal{F}}\left[L_{\mathcal{C}}\left(\boldsymbol{\sigma}(\mu_{\theta^{\text{BL}}}, f(\mathcal{C}))\right)\right]. \tag{15}$$

We use $\boldsymbol{\sigma}(\pi, \mathcal{C})$ to denote the tour generated by policy $\pi$ on instance $\mathcal{C}$. Policy $\mu_{\theta^{\text{BL}}}$ is the deterministic policy induced by $\pi_{\theta^{\text{BL}}}$ by selecting the node with maximum probability.

**Baseline Model Update.** After several policy updates, we would utilize a batch of instances to evaluate the performance of the models. By comparing the average performance of the train model and the baseline model, the baseline model will copy the parameters from the train model if the performance difference is less than a tolerance $\varepsilon$. This implicitly indicates that the baseline model is the best-so-far train model during training.

To evaluate one instance by our model, number of $\omega_{\text{eval}}$ extra instances are augmented using sampled augmentation functions. We use the model to generate $\omega_{\text{eval}} + 1$ tours on the original instance and its augmented instances by deterministic policy. Calculating the tour lengths over the original instance, we select the best tour to be the final solution.

## 5.2 Algorithm for $\text{TS}^4$

Local search methods can further improve the performance of solutions given by constructive methods. We adapt the Monte-Carlo Tree Search (MCTS) based scheme (Fu et al., 2021) into $\text{TS}^3$.

**Heatmap Generation.** To adapt probabilistic sampling in MCTS, we need to define a probability heatmap for the given instance, assigned to every edge belonging to the complete graph. Given a set of solutions $\{\boldsymbol{\sigma}_q\}_{q=0}^{\omega_{\text{eval}}}$ obtained from model evaluation over augmented instances, the heatmap is calculated as:

$$\Pr(e_{ij}) \propto \mathbf{1}_{ij}^{\text{knn}} + \frac{\beta}{\omega_{\text{eval}} + 1} \sum_{q=0}^{\omega_{\text{eval}}} \mathbf{1}_{ij}^{\boldsymbol{\sigma}_q}, \qquad \sum_{j=1}^{n} \Pr(e_{ij}) = 1, \tag{16}$$

where $\mathbf{1}_{ij}^{\text{knn}}$ is an indicator function, indicating whether node $c_j$ is in the top-$k$-th nearest node of $c_i$, and $\mathbf{1}_{ij}^{\boldsymbol{\sigma}_q}$ is also an indicator function indicating whether edge $e_{ij}$ exists in tour $\boldsymbol{\sigma}_q$. We use a weighting parameter $\beta$ to balance the two terms.

Table 1: Comparisons on TSP-Random evaluations by model trained on TSP50 for $\mathrm{TS}^3$.

| Category | TSP-20 | | TSP-50 | | TSP-100 | | TSP-200 | | TSP-500 | | TSP-1000 | |
|---|---|---|---|---|---|---|---|---|---|---|---|---|
| LKH3(100) | 0.13788 s | | 1.78567 s | | 11.34500 s | | 65.01146 s | | 553.47642 s | | 2754.31533 s | |
| Measurements | gap (%) | time (s) | gap (%) | time (s) | gap (%) | time (s) | gap (%) | time (s) | gap (%) | time (s) | gap (%) | time (s) |
| *DIMES | / | / | / | / | / | / | / | / | 14.38 | 0.497 | 14.97 | 1.116 |
| AM | 0.97 | 0.009 | 1.72 | 0.022 | 4.87 | 0.043 | 13.54 | 0.084 | 30.37 | 0.212 | 45.04 | 0.432 |
| POMO | **0.01** | 0.007 | 0.03 | 0.015 | 0.68 | 0.027 | 9.08 | 0.111 | 30.91 | 1.319 | 45.04 | 9.688 |
| TSP Transformer | 0.31 | 0.009 | 0.27 | 0.021 | 2.57 | 0.040 | 14.42 | 0.076 | 43.35 | 0.186 | 68.20 | 0.391 |
| PointerFormer | **0.01** | 0.020 | **0.02** | 0.054 | **0.53** | 0.086 | 8.18 | 0.151 | 19.97 | 1.089 | 27.03 | 7.909 |
| AttGCRN | 0.60 | 0.013 | 28.21 | 0.031 | 50.97 | 0.061 | 56.27 | 0.237 | 85.43 | 0.558 | 118.05 | 1.070 |
| $\mathrm{TS}^3$ | 0.03 | 0.099 | 0.51 | 0.190 | 1.82 | 0.444 | **3.98** | 0.735 | **6.68** | 1.926 | **8.10** | 5.578 |

The first term forces the MCTS algorithm to consider all nodes in the $k$-NNs. The second term represents the frequency of each edge selected by the models among different augmented instances. Intuitively, we construct the heatmap to guide the search algorithm over the reduced action space, based on the preferences by the model.

# 6 EXPERIMENTAL RESULTS

## 6.1 EXPERIMENTAL SETUP

To demonstrate the performance of our method, we design a series of experiments on different datasets and compare the performance with other baselines. Codes can be found in Appendix A.1 for reproduction.

In our experiments, we consider two metrics: the solution gap and the inference time. The solution gap is computed by the average gap to the optimal solutions among all generated instances, and the inference time is computed by the average inference time per instance. Parallelization is not explored, meaning that all methods only proceed with one instance at one run. To ensure the fairness of comparisons, all baseline methods and our method are tested under the same machine (a single Intel Core i7-12700 CPU and a single RTX 3060 GPU). We use the same algorithm used in baseline model update to evaluate instances for $\mathrm{TS}^3$. Full configurations of our settings can be found in Appendix A.3, and it takes about 3 days to train a model following. For some baseline methods marked with *, it means that we directly take the reported data from others' paper.

**Dataset.** Our experiments focus on two datasets: TSP-Random and TSPLIB. TSP-Random consists of instances generated by uniformly sampling a specific number of nodes within the range of $[0, 1]^2$. It includes six sets of TSPs, with different sizes $n = 20, 50, 100, 200, 500, 1000$, each with 1000 instances. We utilize LKH3 (Helsgaun, 2009) to solve those random instances and to calculate the gap. TSPLIB (Reinelt, 1991), as a well-known TSP library that contains different sizes of TSP instances for practial applications, is also included. In our experiments, we consider the same test set as PointerFormer (Jin et al., 2023).

**Baselines.** To get a comparison of generalization ability, all of the models is trained on TSP50 in our experiments. Our baselines include AM (Kool et al., 2019), POMO (Kwon et al., 2020), TSP Transformer (Bresson & Laurent, 2021), PointerFormer (Jin et al., 2023), Att-GCRN (Fu et al., 2021) and DIMES (Qiu et al., 2022), compared with our $\mathrm{TS}^3$. For those models considering multiple decoding strategies, we assume to use multiple optima (if available) or sampling.

## 6.2 PERFORMANCE ANALYSIS

**Performance on TSP-Random.** As shown in Table 1, PointerFormer achieves the best performance when the size of TSP is less than or equal to 100. Compared to PointerFormer, our model, $\mathrm{TS}^3$, achieves close but worse performance on small-sized TSP. However, when applied to large-sized TSP, $\mathrm{TS}^3$ can achieve much better performance with low increase on inference time.

**Performance on TSPLIB.** Table 6.2 demonstrates the overall performance of baseline models and our models on TSPLIB. Results marked with * are those reported by Jin et al. (2023), and the TSPLIB results for Pointer-Former corresponds to Model100, instead of Model50 in TSP-Random. Similar to the results on TSP-Random, our $TS^3$ achieves a slight worse performance on small-sized TSPs while keeping a dominant performance on large-sized TSPs. The detailed results for TSPLIB instances can be checked in Appendix A.4.

Table 2: TSPLIB evaluations.

| Model | 1∼100 | 101∼500 | 501∼1002 |
|---|---|---|---|
| *AM | 15.36% | 78.18% | 139.02% |
| *POMO | 1.20% | 6.99% | 26.93% |
| *PointerFormer | 1.33% | 5.43% | 18.65% |
| $TS^3$ | 2.04% | 4.73% | 8.57% |

## 6.3 ABLATION STUDY AND SENSITIVITY ANALYSIS

**Ablation Study.** As shown in Section 4 and Section 5, our $TS^3$ involves different components and mechanisms to improve the performance. Table 3 gives a view of the effects of using those components and mechanisms. The dramatic decrease of the performance when dropping the components in $TS^2$ reflects their significance, especially for the scale operation and the *local encoder*. For other mechanisms in $TS^3$, it can be observed that the overall performance increases after applying this mechanism.

**Sensitivity Analysis.** Several hyperparameters are involved in our model training, including the number of global layers $\gamma_{\text{global}}$, the number of local layers $\gamma_{\text{local}}$, the number of heads $\gamma_{\text{heads}}$ in the attention layers, the augmentation size $\omega$ during training, the parameter $k$ for the local $k$-NNs, and the size of training instances $n_{\text{train}}$. Following a quick grid search, we set the hyperparameters as follows: $\gamma_{\text{global}} = 4$, $\gamma_{\text{local}} = 6$, $\gamma_{\text{heads}} = 8$, $\omega = 7$, and $k = 12$. However, as suggested by Table 3, the performance of our model is very stable to changes of hyperparameters.

Table 3: Ablation study and sensitivity analysis.

| Hyperparameters | TSP50 | TSP200 | TSP1000 |
|---|---|---|---|
| $TS^3$ | 0.51% | 3.98% | 8.10% |
| w/o Scale | 0.59% | 4.92% | 23.51% |
| w/o Global Encoder | 0.84% | 6.50% | 12.39% |
| w/o Local Encoder | 1.50% | 11.39% | 36.55% |
| w/o Augmentation | 0.61% | 4.65% | 9.95% |
| $\gamma_{\text{local}} = 4$ | 0.63% | 4.24% | 8.99% |
| $\gamma_{\text{global}} = 2$ | 0.61% | 4.19% | 8.63% |
| $\gamma_{\text{heads}} = 4$ | 0.66% | 4.22% | 8.44% |
| $\omega = 15$ | 0.70% | 4.27% | 8.20% |
| $k = 15$ | 0.53% | 4.11% | 8.48% |
| $n_{\text{train}} = 30$ | 0.56% | 4.55% | 10.10% |
| $n_{\text{train}} = 100$ | 0.52% | 2.59% | 7.76% |

## 6.4 ANALYSIS OVER $TS^4$

We evaluate $TS^4$ on the same dataset, with the best solution generated by $TS^3$ as the initial solution of MCTS. $TS^4$-uniform is $TS^4$ with the heatmap replaced by a uniform one. Although our performance is the best shown in the table, we do not conclude that $TS^4$ is dominant to other methods, since we do not include a systematic evaluation nor we also do not include other model-search papers. We just use this to show that our $TS^4$ can at least show a competitive result compared with other SOTA works. Detailed experimental results for $TS^4$ are available in Appendix A.4.

Table 4: TSP-Random evaluations with MCTS.

| Methods | TSP200 | TSP500 | TSP1000 |
|---|---|---|---|
| *DIMES+MCTS | / | 2.64% | 3.98% |
| *DIMES+AS+MCTS | / | 1.76% | 2.46% |
| AttGCRN+MCTS | 0.64% | 2.10% | 2.69% |
| $TS^4$-uniform | 0.10% | 2.91% | 5.39% |
| $TS^4$ | 0.04% | 0.45% | 2.05% |

## 7 CONCLUSION

We demonstrated that approximate invariance can be exploited to improve generalization by proposing an architecture called $TS^2$, training it on small-sized instances with data augmentation to form $TS^3$, and evaluating it on larger instances. Our $TS^3$ achieves a dominant performance among all end-to-end methods considering cross-size generalization. In addition, we also propose a simple but generic method to adapt Monte-Carlo Tree Search. Furthermore, we performed multiple experiments to investigate the effects of our designed components and the sensitivity to the hyperparameters. As future work, we plan to extend our architecture to more complex routing problems, such as Vehicle Routing Problem (VRP).

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

# A  APPENDIX

## A.1  REPRODUCTION

During anonymous submission, we provide our code used for experiments in supplementary materials. To reproduce our work, one can follow the instructions witten in README.md attached in supplementary materials.

Due to size limitations, we do not include our trained model and generated datasets in supplementary materials. All of them will be available through Github after our work being accepted, but the link is hidden during anonymous submission.

## A.2  ALGORITHMS

Algorithm 1 gives the pseudocode for our modified REINFORCE algorithm.

---

**Algorithm 1: Training Algorithm for $\text{TS}^3$**

---

**Require:** epochs $T$, epoch steps $M$, batch size $N$, augmentation size $\omega$, learning rate $\alpha$, tolerance $\varepsilon$, augmentation function $f$
1: Initialize $\theta, \theta^{\text{BL}} \leftarrow \theta$
2: **for** epoch $= 1$ **to** $T$ **do**
3:  **for** step $= 1$ **to** $M$ **do**
4:    $\{\mathcal{C}_{i,0}\}_{i=1}^{N} \leftarrow$ RandomInstance()
5:    $\{\{\mathcal{C}_{i,j}\}_{j=0}^{\omega}\}_{i=1}^{N} \leftarrow \{\mathcal{C}_{i,0}\}_{i=1}^{N} \cup \{\{f(\mathcal{C}_{i,0})\}_{i=1}^{N}\}_{j=1}^{\omega}$
6:    $\boldsymbol{\sigma}_{i,j} \leftarrow$ SampleRollout$(\mathcal{C}_{i,j}, \pi_\theta)$, $\boldsymbol{\sigma}_{i,j}^{\text{BL}} \leftarrow$ GreedyRollout$(\mathcal{C}_{i,j}, \pi_{\theta^{\text{BL}}})$
7:    $G_{ij} \leftarrow L_{\mathcal{C}_{i,0}}(\boldsymbol{\sigma}_{i,j})$, $B_{ij} \leftarrow L_{\mathcal{C}_{i,0}}(\boldsymbol{\sigma}_{i,j}^{\text{BL}})$
8:    $\nabla\mathcal{L}(\theta) \leftarrow \dfrac{1}{N(\omega+1)} \sum_{i,j} (G_{ij} - B_{ij}) \nabla_\theta \log \pi_\theta(a|s)$
9:    $\theta \leftarrow \theta - \alpha\nabla\mathcal{L}(\theta)$
10:   **end for**
11:   **if** Test$(\theta, \theta^{\text{BL}}) < \varepsilon$ **then**
12:     $\theta^{\text{BL}} \leftarrow \theta$
13:   **end if**
14: **end for**

---

Algorithm 2 presents how a new TSP instance can be solved by $\text{TS}^4$.

---

**Algorithm 2: Evaluation Algorithm of $\text{TS}^4$.**

---

**Require:** instance $\mathcal{C}$, model
1: Scale instance $\mathcal{C}$ to $\mathcal{C}_0$.
2: Augment $\mathcal{C}_0$, $\{\mathcal{C}_q\}_{q=1}^{\omega_{\text{eval}}} = \{f(\mathcal{C}_0)\}_{q=1}^{\omega_{\text{eval}}}$.
3: Generate solutions $\{\boldsymbol{\sigma}_q\}_{q=0}^{\omega_{\text{eval}}}$ by model inference.
4: Find the best solution $\boldsymbol{\sigma}^* = \arg\min_{\boldsymbol{\sigma}} [L_{\mathcal{C}_0}(\boldsymbol{\sigma})]$.    ($\leftarrow$ outputs of $\text{TS}^3$)
5: Generate heatmaps $\Pr(e_{ij})$ by solutions $\{\boldsymbol{\sigma}_q\}_{q=1}^{\omega_{\text{eval}}}$.
6: Run MCTS with initial solution $\boldsymbol{\sigma}^*$ and heatmaps $\Pr(e_{ij})$.
7: Find the best solution $\boldsymbol{\sigma}^+$ during MCTS optimization.    ($\leftarrow$ outputs of $\text{TS}^4$)

---

## A.3  HYPERPARAMETERS

Table 5 shows a detailed configuration used in the experiments.

## A.4  EXTRA RESULTS

Table 6 shows the full results for $\text{TS}^4$ and AttGCRN+MCTS on TSP-Random datasets. For these two methods, we impose a soft limit $0.01 \times n$ seconds per instance for both methods. It means that search time for $\text{TS}^4$ is less than AttGCRN+MCTS, due to the fact that $\text{TS}^3$ costs more time than AttGCRN. The reported data in DIMES (Qiu et al., 2022) is about 3 seconds per instance for

Table 5: Hyperparameters.

| Hyperparameter | Notation | Value |
|---|---|---|
| number of global encoder layers | $\gamma_{\text{global}}$ | 4 |
| number of local encoder layers | $\gamma_{\text{local}}$ | 6 |
| number of decoder layers | | 2 |
| number of MHA heads | $\gamma_{\text{heads}}$ | 8 |
| dimension of embeddings | | 128 |
| dimension of feed-forward layers | | 512 |
| TSP size during training | | 50 |
| augmentation size during training | $\omega$ | 7 |
| training epochs | $T$ | 170 |
| steps for each epoch | $M$ | 1600 |
| batch size at each training step | $N$ | 6 |
| learning rate | $\alpha$ | $1.875 \times 10^{-5}$ |
| tolerance of baseline model update | $\varepsilon$ | $10^{-3}$ |
| size of the $k$-NNs / cardinality of the reduced action space | $k$ | 12 |
| range of noisy perturbation | | $10^{-5}$ |
| TSP size during evaluation | $n$ | $[20, 50, 100, 200, 500, 1000]$ |
| augmentation size during evaluation | $\omega_{\text{eval}}$ | $[63, 31, 31, 15, 7, 7]$ |
| scale of heatmap generation | $\beta$ | 64 |
| time limit of TSP inference (for search-based methods) | | $0.01 \times n$ |
| maximum opt for MCTS local move | | $[10, 15, 40, 40, 40, 40]$ |

Table 6: Comparisons on TSP-Random evaluations by model trained on TSP50 for $\text{TS}^4$.

| Category | TSP-20 | | TSP-50 | | TSP-100 | | TSP-200 | | TSP-500 | | TSP-1000 | |
|---|---|---|---|---|---|---|---|---|---|---|---|---|
| LKH(100) | 0.13788 s | | 1.78567 s | | 11.34500 s | | 65.01146 s | | 553.47642 s | | 2754.31533 s | |
| Measurements | gap (%) | time (s) | gap (%) | time (s) | gap (%) | time (s) | gap (%) | time (s) | gap (%) | time (s) | gap (%) | time (s) |
| AttGCRN+MCTS | **0.00** | 0.194 | **0.02** | 0.482 | 0.22 | 0.963 | 0.64 | 2.042 | 2.10 | 5.081 | 2.69 | 10.139 |
| $\text{TS}^4$ | **0.00** | 0.193 | 0.04 | 0.481 | **0.08** | 0.960 | **0.45** | 2.009 | **1.58** | 5.001 | **2.05** | 9.892 |

DIMES+MCTS. However, they utilize 16 threads of CPU and our method is tested only on a single thread. It cannot be concluded which method is better under these settings.

Table 7 shows the full version of TSPLIB evaluation results. All results of comparison methods are data reported in appendix by Jin et al. (2023). For PointerFormer, its results are from taken the colomn Model100. A "/" is used to represent a missing result.

# B ADDITIONAL RELATED WORKS

Deep (Reinforcement) Learning for combinatorial optimization has just been explored for several years, and TSP, one of the most well-known and widely applied NP-hard problems, has become a good evaluator.

There are already several well-performed solvers for TSP. Gurobi (Gurobi Optimization, LLC, 2023) and Concorde (Applegate et al., 2015) are exact solvers which can be used to find optimal solutions. LKH3 (Helsgaun, 2017) is a SOTA heuristic solver, which can output (near-)optimal solutions in a much shorter time compared with Gurobi. However, these solvers are still inefficient in some cases, especially over large-sized instances. This motivates researchers to explore how deep (reinforcement) learning can achieve good performance under fast inference.

### B.1 END-TO-END METHODS

In the early days, researchers tried to find a suitable model for TSP. Ptr-Net (Vinyals et al., 2015) is proposed as a specific model for combinatorial optimization, where the output sequence has the same length as the input sequence. Khalil et al. (2017) mechanism proposes an S2V mechanism to learn graph embeddings. Traditional GNN is explored by Joshi et al. (2019), where a GCN is constructed for learning graph representations. By leveraging advantages from both Ptr-Net and GNN, Ma et al. (2019) develops GPN for TSP. One of the most influential works is AM (Kool et al., 2019) building the model by attention layers from Transformer (Vaswani et al., 2017). This paper lays a solid foundation of attention-based models, and the effectiveness of Transformer-structure on TSP is further evaluated by Bresson & Laurent (2021). Inspired by the Transformer-structured TSP solver, Jin et al. (2023) proposes PointerFormer by extending Ptr-Net to a multi-pointer-network into the Transformer decoder.

Among all the mentioned work above, only a few of them are using supervised learning (Vinyals et al., 2015; Joshi et al., 2019). Many researchers choose reinforcement learning to train the neural network model because it does not need to solve for optimal solutions, and there are also many explored ideas for improving the reinforcement learning algorithm. POMO (Kwon et al., 2020) shows that an RL agent for combinatorial optimization can be drastically improved by multiple optima (i.e. multiple greedy solutions, either generated by different start points or augmented instances). EMAGIC (Ouyang et al., 2021) shows the benefits of introducing invariance to an RL-based TSP solver. Sym-NCO (Kim et al., 2022) further exploits the rotation data augmentation for combinatorial optimizations and develops an RL algorithm to learn invariance. DIMES (Qiu et al., 2022) and Omni-TSP Zhou et al. (2023) adapt meta-RL to train an RL solver generalizable to unseen tasks. H-TSP (Pan et al., 2023) develops a hierarchical two-stage model to construct TSP solutions efficiently. AMDKD (Bi et al., 2022) adapts knowledge distillation to learn a generalizable student model. DROP (Jiang et al., 2022) adapts group distributionally robust optimization to learn from multiple datasets with different data distributions. MVGCL (Jiang et al., 2023) adapts contrastive learning to encode representative features from the graph.

Several papers are working in a quite different and interesting direction, which we call it decoding strategy. Before this field is explored, only several traditional methods are available, including greedy, sampling, and beam search. POMO (Kwon et al., 2020) can also be viewed as a special decoding strategy since multiple optima decoding can be applied to any constructive model. LEHD (Luo et al., 2023) designs its decoder to support complex operations so that the model can iteratively perform local optimizations for generated solutions. AS (Bello et al., 2016) is the first paper introducing learning into the decoding strategy. It allows the model parameters to be slightly tuned on test instances for better evaluation. Inspired by the novel idea from it, EAS (Hottung et al., 2021) and SGBS (Choo et al., 2022) are successively proposed. A valuable property of these papers is the compatibility with the end-to-end methods mentioned above.

### B.2 MODEL-SEARCH METHODS

Since our work mainly focuses on developing a end-to-end methods. We only have a short discussion over model-search methods.

Model-search methods work for introducing deep learning to a local search framework. Most of the work in this realm deals with two components: sub-region scoring and local optimization. Sub-region scoring can be used in two ways: either 1) promising candidates for local moves, or 2) bad regions to be re-created can be selected by scores. Local optimization is performed by 1) evaluating the promising local moves, or 2) repairing the selected regions. Different papers lead to different designs for these two components, and we refer to D2O (da Costa et al., 2021), NeuroLKH (Xin et al., 2021), LCP (Kim et al., 2021), MCTS (Fu et al., 2021), SO (Cheng et al., 2023)... for more details. Among them, MCTS is a special one, where its parametrization is learned during the search. This allows the method to be free from a pre-trained model for local search, which can be adapted to end-to-end methods easily.

Comparing the experimental results of end-to-end methods and model-search methods, it can be observed that model-search methods tend to have a much better performance without considering time consumption. Therefore, many TSP papers equip their own model with MCTS for further improvements after constructing an initial solution. UTSP (Min et al., 2023) learns a transition

matrix processed by a cyclic permutation operation to be the heatmap for MCTS. DIMES (Qiu et al., 2022) uses the continuous parametrization from GNN to be the heatmap of MCTS. DIFUSCO (Sun & Yang, 2023) extends the usage from DIMES and further exploits multiple MCTS runs using multiple sampled heatmaps.

### B.3 More Discussions

**Cross-size Generalization.**    Researchers gradually attach importance to cross-size generalizability (Khalil et al., 2017; Fu et al., 2021; Qiu et al., 2022; Pan et al., 2023; Zhou et al., 2023; Gao et al., 2023). Focus on TSP, this generally means that a solver trained on small-sized TSP instances should be adaptable to large-sized TSP without excess performance loss. We should notice here that some papers (Jin et al., 2023; Sun & Yang, 2023) are considering some different "cross-size generalizability", where they try to develop a method that can be trained on large-sized TSP. In our paper, we just treat them as normal TSP solvers.

**Cross-distribution Generalization.**    Cross-distribution generalizability measures whether a solver can solve instances from multiple data distributions robustly. Though most of the methods on TSP are evaluated on random TSP instances, real-world applications do not strictly follow a uniform distribution, such as TSPLIB (Reinelt, 1991). More attention are paid to this field after a systematic evaluation of mutators for routing problems (Bossek et al., 2019). Some mentioned papers indeed is aimed to exploit the cross-distribution generalizability of an RL-based TSP solver (Jiang et al., 2022; Bi et al., 2022; Jiang et al., 2023; Zhou et al., 2023).

**Usage of $k$-NN.**    There are already some papers utilizing $k$-NN as a trick to perform graph sparsification (Qiu et al., 2022) and reduce computational complexity (Fu et al., 2021), though this is not emphasized in their papers. MVGCL (Jiang et al., 2023), to the best of our knowledge, is the first paper to formally explore the local information induced by a $k$-NN graph. Nonetheless, like the previously mentioned work, they utilize $k$-NN on a complete graph. Different from MVGCL, ELG (Gao et al., 2023) applies $k$-NN during construction in a local policy. As explained in the main article (Section 2), even though our method and ELG both use $k$-NN during construction, our method is essentially different from them with respect to ideas and interpretations.

## C   Implementation Details

### C.1   Augmentation Functions

Given an instance $\mathcal{C}$, it is processed by the following operations. Rotation $\rightarrow$ Scale $\rightarrow$ Reflection $\rightarrow$ Scale $\rightarrow$ Noisy Perturbation $\rightarrow$ Scale.

**Rotation.**    Uniformly sample a random angle $\phi \in [0, 2\pi]$. All nodes are rotated with respect to the origin by the angle $\phi$. Formally, $(x, y) \rightarrow (x\cos\phi - y\sin\phi, x\sin\phi + y\cos\phi)$.

**Reflection.**    Binomially sample a number $r \in \{-1, 1\}$. All nodes are reflected (or not) by multiplying the number to their coordinates. Formally, $(x, y) \rightarrow (rx, ry)$.

**Noisy Perturbation.**    Determine the range of noisy perturbation $\xi_{\max}$. For each node in an instance, it is moved in a relative range of $[-\xi_{\max}/2, \xi_{\max}/2]^2$. Formally, $(x, y) \rightarrow (x + \xi_x, y + \xi_y)$, where $(\xi_x, \xi_y) \in [-\xi_{\max}/2, \xi_{\max}/2]^2$ are uniformly sampled for every node.

**Scale.**    All nodes are then scaled to a unit board $[0, 1]^2$. The scaling factor is set to a number as large as possible. Formally, $(x, y) \rightarrow ((x - x_{\min})/\text{SF}, (y - y_{\min})/\text{SF})$, where SF is the scaling factor equals to $\max_{i,j}\{x_{\max} - x_{\min}, y_{\max} - y_{\min}\}$.

### C.2   Local Move for MCTS

This is the explanation of kopt local move used in MCTS. A kopt local move can be expressed by a sequence of $(u_1, v_1, u_2, v_2, \cdots, u_h, v_h, u_1)$, where $(u_1, v_1), \cdots (\sigma_{u_h}, \sigma_{v_h})$ are broken in the tour $\boldsymbol{\sigma}$,

and $(\sigma_{v_1}, \sigma_{u_2}), \cdots (\sigma_{v_h}, \sigma_{u_1})$ are connected. Notices that $(u_1, v_1, \cdots, u_h, v_h)$ should be different nodes, so the maximum number of $h$ cannot exceed $n/2$. Also, as a local search method, a local move will not be adapted if it cannot contribute to a performance improvement.

**Selection of $u_1$ and $v_1$.** The first index $u_1$ can be selected by a given probability over all nodes, e.g. uniform distribution. As $(u_1, v_1)$ represents an edge to be removed in the tour $\boldsymbol{\sigma}$, we set $v_1 = u_1 + 1$.

**Selection of $u_{i+1}$ given $v_i$ ($i \geq 1$).** Since $(v_i, u_{i+1})$ represents an edge to be connected in the tour $\boldsymbol{\sigma}$, it is forbidden to select $u_{i+1} = v_i \pm 1$. Then, $u_{i+1}$ is sampled based on the heatmaps, so that $(v_i, u_{i+1})$ corresponds to a promising edge for finding near-optimal solutions.

**Selection of $v_i$ given $u_i$ ($i \geq 2$).** Since $(u_i, v_i)$ represents an edge to be removed in the tour $\boldsymbol{\sigma}$, it should be already connected with each other, indicating that $v_i = u_i \pm 1$. The two options yield two kopt local moves $(u_1, v_1, \cdots, u_i, u_i + 1, u_1)$ and $u_1, v_1, \cdots, u_i, u_i - 1, u_1$. The value of $v_i$ is expected to induce a local move resulting in a feasible solution, so that we have the choice to terminate in the next iteration. By geographical analysis, there is exactly one local move (either induced by $(u_i + 1)$ or $(u_i - 1)$, dependent on the previous selections) satisfy this condition. The other local move results in two independent loops in a TSP instance, leading to an infeasible solution. We assign the value which satisfies our condition to $v_i$. This also means that the selection of $v_i$ ($i \geq 2$) is deterministic.

To know the details how MCTS updates the probability heatmaps dynamically, please refer to Fu et al. (2021).

Table 7: TSPLIB results.

| TSPLIB | Opt (len) | $\text{TS}^3$ (%) | *AM (%) | *POMO (%) | *PointerFormer (%) | $\text{TS}^4$ (%) | *DRL-2opt (%) | *AM+LCP (%) |
|---|---|---|---|---|---|---|---|---|
| att48 | 33524 | 0.29 | / | / | / | 0.00 | / | / |
| eil51 | 426 | 0.70 | 2.11 | 1.17 | **0.00** | **0.47** | 1.17 | 0.73 |
| berlin52 | 7542 | 0.36 | 50.3 | **0.00** | **0.00** | **0.03** | 0.07 | 0.10 |
| st70 | 675 | 1.93 | 1.33 | **0.00** | **0.00** | **0.30** | 0.89 | 0.74 |
| eil76 | 538 | 3.90 | 2.23 | **0.00** | **0.00** | **1.12** | 2.04 | 1.64 |
| pr76 | 108159 | 0.89 | 6.60 | **0.00** | 0.89 | 0.89 | 1.05 | **0.44** |
| rat99 | 1211 | **2.39** | 12.55 | 3.47 | 3.88 | **0.66** | 3.39 | 6.67 |
| kroA100 | 21282 | **0.59** | 33.75 | 1.26 | 0.80 | **0.01** | 5.80 | 2.95 |
| kroB100 | 22141 | 3.42 | 25.36 | **0.26** | 1.98 | **0.00** | 6.21 | 1.51 |
| kroC100 | 20749 | 2.31 | 14.00 | 1.29 | **1.01** | **0.01** | 0.52 | 2.84 |
| kroD100 | 20749 | 6.25 | 26.63 | **5.51** | 5.66 | 2.63 | 6.25 | **1.97** |
| kroE100 | 22068 | 2.08 | 6.10 | 1.40 | **1.33** | **0.00** | 1.29 | 1.90 |
| rd100 | 7910 | 1.38 | 3.35 | **0.00** | **0.00** | **0.00** | 0.54 | 1.90 |
| eil101 | 629 | 3.34 | 2.23 | **0.16** | **0.16** | 1.75 | **1.59** | 2.59 |
| lin105 | 14379 | 2.01 | 122.71 | **1.40** | 2.75 | **0.03** | 12.39 | 3.86 |
| pr107 | 44303 | **0.65** | 5.55 | 1.36 | 2.27 | **0.00** | 15.52 | / |
| pr124 | 59030 | 2.91 | 3.88 | **0.08** | **0.08** | **0.00** | 1.39 | 3.84 |
| bier127 | 118282 | **1.57** | 14.44 | 5.28 | 2.34 | **0.75** | 5.32 | 8.92 |
| ch130 | 6110 | 1.98 | 3.62 | 0.23 | **0.10** | 0.33 | 1.55 | 0.57 |
| pr136 | 96772 | 1.99 | 5.76 | **0.75** | 0.98 | **0.00** | 2.90 | 1.56 |
| pr144 | 58537 | 2.66 | 10.01 | 0.66 | **0.17** | 0.05 | 3.37 | 3.47 |
| ch150 | 6528 | 2.99 | 4.76 | **0.46** | 0.52 | 0.05 | 1.38 | / |
| kroA150 | 26524 | 1.36 | 19.14 | **0.79** | 3.34 | **0.00** | 7.07 | 3.68 |
| kroB150 | 26130 | 3.32 | 22.63 | **1.73** | 2.60 | **0.04** | 6.50 | 3.18 |
| pr152 | 73682 | 10.02 | 15.06 | **0.95** | 1.26 | **0.00** | 6.72 | 1.52 |
| u159 | 42080 | **0.37** | 8.56 | 1.02 | 0.99 | **0.00** | 1.71 | 10.84 |
| rat195 | 2323 | **3.01** | 18.94 | 9.73 | 7.10 | **0.73** | 5.68 | 10.81 |
| d198 | 15780 | 7.23 | 440.06 | 18.92 | 15.91 | **0.20** | 32.97 | / |
| kroA200 | 29368 | **1.91** | 31.91 | 1.94 | 5.23 | **0.18** | 9.67 | 6.14 |
| kroB200 | 29437 | **3.50** | 29.45 | 3.69 | 5.23 | **0.21** | 6.74 | / |
| ts225 | 126643 | **5.06** | 11.15 | 7.56 | 6.22 | **3.36** | 4.76 | 6.46 |
| tsp225 | 3916 | **3.88** | 34.04 | 5.77 | 9.01 | **0.00** | 7.87 | 14.50 |
| pr226 | 80369 | 4.10 | 15.90 | **2.82** | 1.52 | **0.11** | 12.00 | 6.09 |
| gil262 | 2378 | 5.51 | 13.37 | **2.61** | 1.98 | **0.88** | 4.12 | 5.49 |
| pr264 | 49135 | 8.40 | 39.77 | 11.94 | **4.80** | **0.00** | 35.82 | / |
| a280 | 2579 | **5.54** | 29.12 | 10.00 | 9.50 | **0.70** | 8.88 | / |
| pr299 | 48191 | **4.13** | 432.65 | 11.54 | 13.54 | **0.23** | 26.88 | / |
| lin318 | 42029 | **5.38** | 22.55 | 8.08 | 5.43 | **0.75** | 12.91 | 10.72 |
| rd400 | 15281 | **6.19** | 24.40 | 11.62 | 9.02 | **1.14** | 11.31 | 8.10 |
| fl417 | 11861 | 15.10 | 117.63 | 14.06 | **7.76** | **0.55** | 48.85 | / |
| pr439 | 107217 | **7.49** | 260.14 | 14.27 | 13.34 | **5.27** | 41.21 | 22.18 |
| pcb442 | 50778 | **5.74** | 33.68 | 14.79 | 11.13 | **0.64** | 16.70 | 12.35 |
| d493 | 35002 | **14.58** | 552.35 | 45.53 | 18.57 | **1.40** | 61.78 | / |
| u574 | 36905 | **7.88** | 105.29 | 22.90 | 17.92 | **1.60** | 8.92 | / |
| rat575 | 6773 | **5.65** | 53.37 | 22.87 | 17.98 | **1.28** | 26.07 | / |
| p654 | 34643 | 12.25 | 239.12 | 21.17 | **11.03** | 0.27 | 59.84 | / |
| d657 | 48912 | **9.87** | 325.89 | 32.46 | 18.81 | **1.50** | 60.38 | / |
| u724 | 41910 | **5.28** | 89.79 | 26.10 | 20.49 | **1.21** | 51.50 | / |
| rat783 | 8806 | **6.92** | 75.24 | 27.66 | 22.23 | **2.09** | 68.24 | / |
| pr1002 | 259045 | **12.16** | 84.42 | 35.34 | 22.07 | **7.75** | 23.04 | / |
| vm1084 | 239297 | 9.69 | / | / | / | 5.02 | / | / |
| pcb1173 | 56892 | 8.95 | / | / | / | 1.51 | / | / |
| d1291 | 50801 | 8.76 | / | / | / | 2.21 | / | / |
| rl1304 | 252948 | 8.60 | / | / | / | 5.90 | / | / |
| rl1323 | 270199 | 12.63 | / | / | / | 6.90 | / | / |
| nrw1379 | 56638 | 7.23 | / | / | / | 1.75 | / | / |
| fl1400 | 20127 | 13.33 | / | / | / | 1.47 | / | / |
| u1432 | 152970 | 6.49 | / | / | / | 5.42 | / | / |
| fl1577 | 22249 | 13.84 | / | / | / | 1.20 | / | / |
| d1655 | 62128 | 12.70 | / | / | / | 2.31 | / | / |
| vm1748 | 336556 | 10.78 | / | / | / | 6.86 | / | / |
| u1817 | 57201 | 10.56 | / | / | / | 3.16 | / | / |
| rl1889 | 316536 | 12.56 | / | / | / | 6.53 | / | / |
| d2103 | 80450 | 9.04 | / | / | / | 0.37 | / | / |
| u2152 | 64253 | 11.05 | / | / | / | 3.00 | / | / |
| u2319 | 234256 | 1.91 | / | / | / | 1.35 | / | / |
| pr2392 | 378032 | 12.49 | / | / | / | 6.90 | / | / |
| pcb3038 | 137694 | 9.82 | / | / | / | 7.02 | / | / |
| fl3795 | 28772 | 20.38 | / | / | / | 3.91 | / | / |
| fnl4461 | 182566 | 9.03 | / | / | / | 6.35 | / | / |
| rl5915 | 565530 | 14.47 | / | / | / | 7.30 | / | / |
| rl5934 | 556045 | 16.34 | / | / | / | 7.95 | / | / |

