# OpenReview forum: "Generalizable Deep RL-Based TSP Solver via Approximate Invariance"
_ICLR.cc/2024/Conference — Submitted to ICLR 2024_

### Official Review · Reviewer_dYNB · 2023-10-16

**Soundness:** 3 good
**Presentation:** 2 fair
**Contribution:** 2 fair
**Rating:** 5
**Confidence:** 4

**Summary:**

This paper studies the generalization of neural TSP solvers. The authors propose a Transformer Structured TSP Solver, which reduces the space of decision-making to the K-nearest neighbors of the current node. By further combining with data augmentation and MCTS, the proposed method (i.e., $TS^4$) could achieve superior generalization performance on large-scale TSP instances when only trained on TSP50.

**Strengths:**

* The motivation is clear. The studied topic (i.e., generalization) is important.
* The proposed method is sound to me. The authors claim three contributions, including (1) the local KNN view, (2) the modified training algorithm with data augmentations, and (3) the combination with MCTS.
* The empirical results look good. The source code is provided.

**Weaknesses:**

* The scope of this paper is limited to TSP. Popular attention-based models (e.g., [1, 2]) could solve a wide range of VRP variants. It is suggested to (at least) include CVRP.

* Novelty:

  * I like the third contribution. Typically, MCTS mainly works with heat-map-based methods [3]. This paper demonstrates the potential of attention-based solvers [1, 2] with MCTS.
  * While the first two seem to be incremental since previous works have already explored them (e.g., [4]).

* The related work and baselines are too limited. This paper studies the generalization issue of DRL-based TSP solvers. Based on the ICLR policy, a comprehensive review and experimental comparison of the recent generalization studies (before June, 2023) is expected. Moreover, Concorde should be added in Table 1, and the gap should be computed w.r.t. its result.

* The justification for Figure 1 is weak.

  * Are you using instances following the uniform distribution? Does the conclusion or empirical observation still hold for other cases, e.g., instances with several clusters of customer nodes?

  * I somewhat don't agree with the conclusion of the right panel of Figure 1 (see below). Recent studies find that the adversarial perturbations [5, 6] may significantly change the optimal solutions of TSP instances. What if the added random perturbation coincides with the adversarial one?

    > "Small perturbations introduce small gaps, which can be regarded as exploiting approximate invariance."

* The problem formulation of TSP (with DRL) should be simple and concise, while the notations of this paper are wordy, making it unclear to the readers.

* Minor:

  * P3 of Introduction: "how (exact) invariance" -> should be approximate?
  * Report the total inference time on a test dataset is better.

[1] Attention, learn to solve routing problems! In ICLR 2019.
[2] POMO: Policy optimization with multiple optima for reinforcement learning. In NeurIPS 2020.
[3] Generalize a small pre-trained model to arbitrarily large TSP instances. In AAAI 2021.
[4] Sym-NCO: Leveraging Symmetricity for Neural Combinatorial Optimization. In NeurIPS 2022.
[5] Generalization of neural combinatorial solvers through the lens of adversarial robustness. In ICLR 2022.
[6] ROCO: A General Framework for Evaluating Robustness of Combinatorial Optimization Solvers on Graphs. In ICLR 2023.

----

**Given all,** I vote for rejection. This paper needs a bit more work before being accepted.

**Questions:**

* The model does not have a normalization layer after the MHA and FF layers. Any explanations?
* For the baseline function, why not just use a simple average over the objective values of all augmented instances, as done in [2]?
* Is the proposed method empirically effective for cross-distribution generalization setting?
* What is the training complexity (e.g., time and GPU memory) of the proposed method, compared with baselines?

---

> ### Author Response · Authors · 2023-11-22
> **Review Response**
>
> - weaknesses
>     - (1) Please check (3) in Point c) in our general response.
>     - (2) Please check Point a) in our general response.
>     - (3) Please check Point b) in our general response.
>       Also, Concorde (or Gurobi) is definitely a better heuristic solver for small-sized TSP instances than LKH3, but the relative performances and the final conclusions do not change whether we select Concorde (or Gurobi) or LKH3 as our "optimality" baselines.
>     - The observation we made about the kNN distribution in Fig.1 holds for other distributions than the uniform distribution, which may explain our good performance on TSPLib.
>       In contrast to the two mentioned papers, we apply a non-adversarial noise. In that case, on average our perturbed instances have similar or the same solutions, which is the property that we want to exploit with approximate invariance.
>
> - questions
>     - _q1: "The model does not have a normalization layer after the MHA and FF layers. Any explanations?"_
>
>       We follow the standard Transformer architecture and includes normalization layer after the MHA and FF layers. We omit it since we regard it as implementation details. We will clarify in the final version.
>
>     - _q2: "For the baseline function, why not just use a simple average over the objective values of all augmented instances, as done in POMO?"_
>
>       I think you are asking about the baseline function update, where we use this technique to compare the performance of the trained model and the baseline model. Our baseline simply follows how the trained policy is used: for a given instance, the final solution is the best found among the solutions generated for the original instance and some perturbed instances.
>
>     - _q3: "Is the proposed method empirically effective for cross-distribution generalization setting?"_
>
>       Please check (2) in Point c) in our general responses.
>
>     - _q4: "What is the training complexity (e.g., time and GPU memory) of the proposed method, compared with baselines?"_
>
>       The time complexity of a single inference procedure is $O(N (\omega+1) n^2 \gamma_{\text{heads}} d_{\text{emb}}^2 k)$ and the memory complexity is $O(N (\omega+1) n \gamma_{\text{heads}} d_{\text{emb}}^2 k)$, where $N$ denotes the batch size, $\omega$ denotes the augmentation size, $n$ denotes the size of TSP, $\gamma_{\text{heads}}$ denotes the number of heads, $d_{\text{emb}}$ denotes the number of dimensions for embedding and $k$ denotes the size of the kNNs. Therefore, the overall time and memory complexity for training is $O(T M n^2 \gamma_{\text{heads}} d_{\text{emb}}^2 k)$ and $O(n \gamma_{\text{heads}} d_{\text{emb}}^2 k)$, where $T$ denotes number of epochs and $M$ denotes steps within one epoch. Since our model is trained on small-sized instances, those costs are affordable. For inference on larger-size TSPs, it is unavoidable to have $O (n^2)$ in the complexity when applying a constructive method involves global information aggregation. In our experiments, we take $N=6$ and $\omega=7$ for training models with TSP-50 on a single RTX 3060 GPU, which has 12GB memory. It takes about 3 days to finish training with $T=170$ and $M=1600$.

---

> > ### Comment · Reviewer_dYNB · 2023-11-23
> > **Reply to Rebuttal**
> >
> > Thanks for your response.
> >
> > * For the presentation issue, it seems that the author does not update the draft.
> >
> > * The empirical evidence wrt W.4 is not provided as well.
> >
> > I have also gone through other reviews, and tend to keep my current rating.

---

> ### Author Response · Authors · 2023-11-23
> **Reply to remind**
>
> Thanks for your kind remind.
> - We have updated our draft for the presentation issue.
> - Due to the limit time, the experiments on the empirical evidence wrt W.4 are still running. We would provide it in the final draft.

---

### Official Review · Reviewer_tWCL · 2023-10-22

**Soundness:** 2 fair
**Presentation:** 2 fair
**Contribution:** 1 poor
**Rating:** 1
**Confidence:** 5

**Summary:**

In this paper, the authors extended the construction type of Transformer-based TSP solver for tackling large instances unseen during training, by coupling the original global view with a kNN based local view. They also extend such Transformer into the Heatmap based solver, which couples with the Monte-Carlo Tree Search. The proposed methods are evaluated on both synthetic instances and benchmark instances (TSPLib).

**Strengths:**

1- The method itself looks reasonable.
2- The paper is easy to follow.

**Weaknesses:**

While the paper is well motivated, its literature review and experimental comparison are extremely poor, which significantly diminishes the novelty and contribution.

1- Regarding the 'generalizable', it missed,
    Towards Omni-generalizable Neural Methods for Vehicle Routing Problems, ICML 2023;
    Learning Generalizable Models for Vehicle Routing Problems via Knowledge Distillation, NeurIPS 2022;
2- Regarding combining Global embedding and Local embedding, it missed,
    Towards Generalizable Neural Solvers for Vehicle Routing Problems via Ensemble with Transferrable Local Policy, Arxiv 2023;
    Multi-View Graph Contrastive Learning for Solving Vehicle Routing Problems, UAI 2023.
3- Regarding comparison on instances with >= 1000 nodes, it missed,
    DIFUSCO: Graph-based Diffusion Solvers for Combinatorial Optimization, Arxiv 2023;
    Unsupervised Learning for Solving the Travelling Salesman Problem, Arxiv 2023;
    DIMES: A Differentiable Meta Solver for Combinatorial Optimization Problems, NeurIPS 2022;
    NeuroLKH: Combining Deep Learning Model with Lin-Kernighan-Helsgaun Heuristic for Solving the Traveling Salesman Problem, NeurIPS 2021;
    Select and Optimize: Learning to solve large-scale TSP instances, AISTATS 2023.

Particularly, the main ideas of this submission is quite similar to some of them. Moreover, it seems the proposed approach is limited to TSP, especially when it is further converted to the heatmap based paradigm, which considerably harmed its applicability to different VRPs. Overall, I think this submission is far away from the standard of an ICLR publication.

**Questions:**

Please refer to the Weaknesses above.

---

> ### Author Response · Authors · 2023-11-22
> **Review Response**
>
> Please check Point b) in our general response.

---

### Official Review · Reviewer_xEdU · 2023-10-30

**Soundness:** 2 fair
**Presentation:** 3 good
**Contribution:** 1 poor
**Rating:** 3
**Confidence:** 4

**Summary:**

The authors present an attention-based approach that solves the TSP using RL based on a well-known existing approach that can improve generalization to larger problem sizes than the ones trained on. They extend the architecture by including additional encodings of the nearest neighbors of the current last node during the generation of the tour. Additionally, they use data augmentation and exploiting invariance and MCTS. However, these techniques are not novel and have been applied to TSP.

**Strengths:**

S1 The approach can achieve SOTA performance in a specific setting.

**Weaknesses:**

W1 Two of the three primary contributions in this paper have been previously explored and are not considered novel (augmentation/invariance, MCTS). The remaining contribution, the encoding of the nearest neighbors, seems relatively straightforward.

W2 The experimental evaluation has several shortcomings. First, the generalization is only evaluated with instances trained on the TSP of size 50. It would be valuable to explore the model's performance across a broader range of problem sizes to gain a more comprehensive understanding of its capabilities.

W3 The paper imposes time limits on several close competitors, which raises questions about whether the proposed method is genuinely superior or simply optimized for a specific time constraint. A more thorough examination without the time limit would be valuable.

W4 Most of the ablation study is done with ts3, but it should be done with ts4 (the complete approach including MCTS).

* Furthermore, the ability of the method to achieve SOTA performance in the tsplib evaluation remains less clear (see appendix), especially since the closest competitor in the random tsp (AttGRCN+MCTS) is missing from the comparison.

**Questions:**

Q1. How would be the generalization look like if the approach is not only trained on size 50?

Q2. How would the competitors perform without the time limit?

---

> ### Author Response · Authors · 2023-11-22
>
> - weaknesses
>   - (1) About contribution and novelty. Please check Points a) and b) of our general response.
>   - (2) About experiments. We have trained our model on TSP-30 and TSP-100 (see table below). The performance increases when trained on larger instances. Interestingly, even training on TSP30 achieves very competitive generalization capabilities.
>
>     |                     | **TSP20** | **TSP50** | **TSP100** | **TSP200** | **TSP500** | **TSP1000** |
>     | ------------------- | --------- | --------- | ---------- | ---------- | ---------- | ----------- |
>     | $\texttt{TS}^3$-30  |    $0.01$%       |    $0.56$%       |       $2.02$%     |      $4.55$%      |    $7.58$%        |        $10.10$%     |
>     | $\texttt{TS}^3$-50  |     $0.03$%      |     $0.51$%      |     $1.82$%       |     $3.98$%       |       $6.68$%     |     $8.10$%        |
>     | $\texttt{TS}^3$-100 |     $0.04$%      |    $0.52$%        |     $1.02$%      |     $2.59$%       |      $5.25$%      |    $7.76$%         |
>   - (3) About time limit. In our experiments, the other baselines given more time are still not competitive with our methods.
>   - (4) About ablation study. Since our main contribution is $\texttt{TS}^3$, we did an extensive ablation study of this architecture. Regarding $\texttt{TS}^4$, to demonstrate the quality of what is learned by $\texttt{TS}^3$, we compare MCTS using our proposed heatmap (i.e., $\texttt{TS}^4$) and MCTS with uniform heatmap. Table 3 shows that the former outperforms the latter.
>   - (5) About comparisons in TSPLIB. We have done a comparison that takes the average of TSP results within a given range as PointerFormer has done. AttGCRN+MCTS results are missing since their code implementation is hard to apply on variable-sized TSPs.
>
> - questions
>   - _q1: "How would be the generalization look like if the approach is not only trained on size 50?"_
>
>     Please check our response to point (2).
>
>   - _q2: "How would the competitors perform without the time limit?"_
>
>     Please check our response to point (3).

---

### Official Review · Reviewer_jwE3 · 2023-10-30

**Soundness:** 3 good
**Presentation:** 2 fair
**Contribution:** 2 fair
**Rating:** 3
**Confidence:** 3

**Summary:**

The paper attempts to solve the generalization issue in TSP. Local and global policies are learned concurrently and the data augmentation is used to enhance the performance of TSP solver. In experiments, the proposed approaches show inferior to some compared approaches on small problem sizes and gain better generalization results on large problem sizes.

**Strengths:**

Generalization issue in TSP is a critical topic that relates to  large problem solving in real-world settings. This work solves TSP from the local and global views to enhance the generalization of TSP solver. At the same time, data augmentation is a commonly used approach to enhance the performance of neural networks and it is empirically effective to make the TSP solver more generalizable.

**Weaknesses:**

The related work is insufficient with too many highly related works not even mentioned. Many literature have applied different approaches to overcome the generalization issue for vehicle routing problems. Please refer to and discuss the highly related ones like
"Towards Generalizable Neural Solvers for Vehicle Routing Problems via Ensemble with Transferrable Local Policy" and "Towards Omni-generalizable Neural Methods for Vehicle Routing Problems".

The compared approaches are too old and can not represent the current best performance for TSP. More recent methods should ought to be compared to render the results more convincing. For instance, the aforementioned two papers already tackled large routing problems with thousands of nodes and achieved small gaps. A comparison to them is recommended to show the reliability of the approach here.

The data augmentation and different baseline functions have been widely studied in literature, degrading the significance of this work. The local and global policies are seen in aforementioned paper. The total novelty of this work is not noticeable. In this sense, more comparisons with recent approaches to fully validate the performance is important, which I think is inferior in this paper.

**Questions:**

1. What if POMO or PointerFormer are trained with data augmentation like rotations or scalings,  which are not special and may be easily applied for these baselines?
2. Since different techniques are added to enhance the generalization. Would these extra techniques obviously raise the training time? How long would it take to train a TS3 model?
3. I don't see any simple ways to extend the approaches to other vehicle routing problems. The authors may explain a bit how the data augmentation and neural network are to be applied to CVRP.

---

> ### Author Response · Authors · 2023-11-22
> **Review Response**
>
> - weaknesses
>   - (1) About related works. Please check [1] and [5] in Point b) of our general response.
>   - (2) About comparison to other SOTA approaches. Among learn to construct method, we have not found any other method that can perform better in terms of cross-size generalization than $\texttt{TS}^3$.
>   - (3) About contribution and novelty. Please check Points a) and b) of our general response.
>
> - questions
>   - _q1: "What if POMO or PointerFormer are trained with data augmentation like rotations or scalings, which are not special and may be easily applied for these baselines?"_
>
>     Note that POMO and PointerFormer train independently over a batch of an instance and its augmented ones (using rotations), while our training method is designed to enforce approximate invariance (using noise+rotation+scaling), i.e., the loss of the augmented instances is computed with respect to the original one. We believe that our training technique if applied in POMO or PointerFormer would improve their performance.
>
>   - _q2: "Since different techniques are added to enhance the generalization. Would these extra techniques obviously raise the training time? How long would it take to train a $\texttt{TS}^3$?"_
>
>     Data augmentation won't cost too much time since it only involves simple operations like noise, rotation, and reflection, which can be processed in batches. The design of the local and global architecture introduces additional training and inference time. However, those costs are affordable since our model can be trained on small instances with low batch size and still achieve a competitive performance. Currently, our models are trained on a single RTX 3060 GPU card within three days.
>
>   - _q3: "I don’t see any simple ways to extend the approaches to other vehicle routing problems. The authors may explain a bit how the data augmentation and neural network are to be applied to CVRP."_
>
>     Please check (3) in Point c) of our general response

---

### Author Response · Authors · 2023-11-22
**General Response (part 1)**

We would like to thank all the reviewers for their efforts in reviewing our paper and providing us with helpful feedback. We provide answers to some commonly raised questions and limitations.

**Point a)**: First of all, we would like to highlight again our main goal and contributions. Our main goal was to demonstrate how approximate invariance can improve cross-size generalization for solving TSP. To the best of our knowledge, approximate invariance is a novel concept for designing generalizable deep RL solvers. To achieve this goal, we propose a constructive RL-based method $\texttt{TS}^3}:
  - (1) a novel Transformer-based architecture combining local and global information enforcing approximate invariance in the policy (using kNN).
  - (2) a novel training method exploiting approximate invariance (using noisy perturbation).
  - (3) In addition, to demonstrate the quality of TS3, we formulate a generic approach to derive a heatmap from a constructive method to be used in MCTS.

---

### Author Response · Authors · 2023-11-22
**General Response (part 2)**

**Point b)**: Related Work: Thank you for sharing these recent references, some of which we were not aware of [3,4,8]. Note that given the ICLR policy (see last question of reviewer FAQ), we are excused not mentioning the most recent ones (officially published after May 28, 2023, [2,5,6,12]) and we are not required to discuss the arXiv-only papers ([1,7]). Since our main focus is to design a learn-to-construct method for TSP, we did not fully discuss some other related work (learn-to-search [11] or cross-distribution generalization [9,10] papers).
Having said that, we take this opportunity to explain how our proposition is different from all those papers. We have updated the related work in our submission. In addition, for the final version, we will present some new experimental results to compare with the most relevant work, which we have not obtained yet due to the limited time of the rebuttal phase.

[1] ELG (Arxiv only) --- _Towards Generalizable Neural Solvers for Vehicle Routing Problems via Ensemble with Transferrable Local Policy._ Indeed, both ELG and our method exploit a local view based on a kNN neighborhood. While ELG combines the outputs of a local policy and a global policy, we only use one local policy, which uses a local view augmented with some global information. This design enforces a kNN approximate invariance over the policy (as justified by our statistical observation in Fig.1), and the aggregation of the local and global views is performed in the embedding space. Since their results on TSPLib are not quite clear and they haven't performed experiments on random TSP dataset, we could not compare our performance.

[2] MVGCL (after May 28, 2023) --- _Multi-View Graph Contrastive Learning for Solving Vehicle Routing Problems._ This method uses kNN on the whole graph for learning representative features by contrastive learning. In contrast, ours uses KNN on the graph induced by the unvisited nodes for RL training. Again, this paper does not exploit kNN approximate invariance of the policy. In addition, this paper also does not focus on cross-size distribution.

[3] Sym-NCO (NeurIPS 2022) --- _Leveraging Symmetricity for Neural Combinatorial Optimization._ Our methods share some similar definitions of the losses (REINFORCE and solution invariance). However, we apply this approach to enforce approximate invariance (e.g., random noise+symmetry), which means that there is no strict guarantee that the perturbed instances and the original instance share the same optimal solutions. Therefore, in contrast to Sym-NCO, we compute the loss of the perturbed instances on the original instance.

[4] DIMES (NeurIPS 2022) --- _Dimes: A differentiable meta solver for combinatorial optimization problems._ Both of our methods focus on cross-size generalization. DIMES trains a GNN via meta-RL on a distribution over TSP instance sizes, while $\texttt{TS}^3$ is trained on a fixed TSP instance size using approximate invariance. Compared with their reported results, $\texttt{TS}^3$ outperforms their trained GNN (sampling) by 6.7% on TSP500 and 5.56% on TSP1000. When combined with MCTS, our method seems to be at least as good.

[5] Omni-VRP (after May 28, 2023) --- _Towards Omni-generalizable Neural Methods for Vehicle Routing Problems._ This paper shares a similar idea to DIMES [4] but extends the meta-RL approach to the CVRP problem. Therefore, it has the same differences to our proposition.

[6] DIFUSCO (after May 28, 2023) --- _DIFUSCO: Graph-based Diffusion Solvers for Combinatorial Optimization._ and
[7] UTSP (Arxiv only) --- _Unsupervised Learning for Solving the Travelling Salesman Problem._
Those propose different methods for learning a heatmap for MCTS. They do not propose a constructive method like us.

[8] EAS (ICLR 2022) --- _Efficient Active Search for Combinatorial Optimization Problems._ This paper proposes a novel decoding strategy that enables the model to actively update during multi-solution sampling. This proposition is orthogonal to our work, which could integrate it.

[9] AMDKD (NeurIPS 2022) --- _Learning generalizable models for vehicle routing problems via knowledge distillation._ and
[10] DROP (AAAI 2022) --- _Learning to Solve Routing Problems via Distributionally Robust Optimization._
In contrast to our focus on cross-size generalization, those two works aim at cross-distribution generalization using techniques different from ours.

[11] NeuroLKH (NeurIPS 2021) --- _NeuroLKH: Combining deep learning model with LinKernighan-Helsgaun heuristic for solving the traveling salesman problem._ and
[12] SO --- _Select and optimize: Learning to solve large-scale tsp instances._
Those two methods are learning to search methods, while we focus on learning to construct methods.

---

### Author Response · Authors · 2023-11-22
**General Response (part 3)**

**Point c)**: Generalization: Indeed, there are various types of generalization:
- (1) Cross-size Generalization --- Our paper focuses on cross-size generalization. Since we train our model on very small instances, TSP50, in contrast to other work, we believe that evaluating on instances from TSP50 to TSP1000 is sufficient to show its cross-size generalizability. For the final version, we can perform an evaluation on larger instances.
- (2) Cross-distribution Generalization --- We did not include a systematic evaluation over non-uniformly-distributed instances. However, TSPLib can be viewed as a dataset for both cross-size and cross-distribution generalization. Here, our model can still achieve a good result, demonstrating cross-distribution capabilities.
- (3) Cross-problem Generalization --- Since TSP is an important problem, we think that focusing only on it is acceptable (see other recently-published work at, e.g., AAAI'23). However, we believe our approach in designing $\texttt{TS}^3$ could be further extended to CVRP, which is our current research effort. Our main ideas can be directly applied or adapted. For instance, our global and local architecture design can still be applied to CVRP by using kNN or an extension taking into account demands to define a suitable notion of approximate invariance over policies. In addition, our data augmentation method is naturally still suitable in CVRP, with potentially additional transformations (e.g., perturbations of demands).

---

### Author Response · Authors · 2023-11-23
**Update on the draft**

Thanks again for the efforts in reviewing our paper. We have updated our draft. The update is as follows:
- More related works are included and presented with a brief discussion.
- The content of the paper has been reorganized to emphasize the core idea.
- Some experimental results haven been included according to the review.

---

### Meta-Review · Area_Chair_75DQ · 2023-12-06

**Metareview:**

Summary: The submission proposes a deep reinforcement learning method for the traveling salesman problem. It claims to obtain better generalizability to larger problem instances than were used during training compared to previous methods by encouraging invariances via a modified policy gradient algorithm enhanced with data augmentation.

+ The paper studies an important problem that would be of interest to the ML community.
+ The proposed method is sound.
+ The experimental results are good in certain settings.

- The paper appears to lack clarity.
- The technical novelty with respect to related work is unclear.

**Justification For Why Not Higher Score:**

During the initial phase, there were concerns raised regarding the technical novelty of the paper. The authors' rebuttal rightly pointed out that the ICLR guidelines imply that recent work and arxiv submissions should not be considered part of the literature. This was taken into account by the reviewers in subsequent considerations. However, the issue of technical novelty with respect to published work from 2022 and earlier still remains an issue for them.

Some of the reviewers also commented on the lack of clarity of the paper, including the new submitted version. Revisions of the work should take into account the detailed comments of the reviewers.

Finally, there are some missing experiments which could not be completed on time (see comments by reviewer dYNB).

**Justification For Why Not Lower Score:**

N/A

---

### Decision · Program_Chairs · 2024-01-16

Reject